# The development and evaluation of a mHealth, community education and navigation intervention to improve clinical breast examination uptake in Segamat Malaysia: A randomised controlled trial

Désirée Schliemann[1], Aminatul Saadiah Abdul Jamil[2,3], Devi Mohan[4], Min Min Tan[2,4], Christopher R. Cardwell[1], Roshidi Ismail[2,4], Nur Aishah Taib[5], Tin Tin Su[2,4‡], Michael Donnelly[1‡]*

1 Centre for Public Health and UKCRC Centre of Excellence for Public Health, Queen's University Belfast, Belfast, United Kingdom, 2 South East Asia Community Observatory (SEACO), Jeffrey Cheah School of Medicine and Health Sciences, Monash University Malaysia, Subang Jaya, Malaysia, 3 Faculty of Science and Technology, Health Industry Technology, Islamic Science University of Malaysia, Nilai, Malaysia, 4 Global Public Health, Jeffrey Cheah School of Medicine and Health Sciences, Monash University Malaysia, Subang Jaya, Malaysia, 5 Department of Surgery, Faculty of Medicine, UM Cancer Research Institute, University of Malaya, Kuala Lumpur, Malaysia

‡ TTS and MD are Joint Senior Authors in this work
* michael.donnelly@qub.ac.uk

## Abstract

### Introduction

Breast cancer (BC) screening uptake in Malaysia is low and a high number of cases present at a late stage. Community navigation and mobile health (mHealth) may increase screening attendance, particularly by women from rural communities. This randomized controlled study evaluated an intervention that used mHealth and community health workers to educate women about BC screening and navigate them to clinical breast examination (CBE) services in the context of the COVID-19 pandemic.

### Methods

Women aged 40–74 years, from Segamat, Malaysia, with a mobile phone number, who participated in the South East Asian Community Observatory health survey, (2018) were randomized to an intervention (IG) or comparison group (CG). The IG received a multi-component mHealth intervention, i.e. information about BC was provided through a website, and telephone calls and text messages from community health workers (CHWs) were used to raise BC awareness and navigate women to CBE services. The CG received no intervention other than the usual option to access opportunistic screening. Regression analyses were conducted to investigate between-group differences over time in uptake of screening and variable influences on CBE screening participation.

**Data Availability Statement:** Data are available from the SEACO (contact via mum. seaco@monash.edu) for researchers who meet the criteria for access to confidential data.

**Funding:** This study is funded by MRC UK (Ref: 537084059) and MIGHT (Ref: 2500235-122-00). The grant application was subject to peer-review by individual academic reviewers and the final decision about funding was made by an expert panel. The funders had no role in study design, data collection and analysis, decision to publish, or preparation of the manuscript.

**Competing interests:** The authors have declared that no competing interests exist.

**Abbreviations:** BC, Breast Cancer; BCAC, Be Cancer Alert Campaign; BCSS, Breast Cancer Support Society Segamat; CBE, Clinical Breast Examination; CG, Comparator Group; CHW, Community Health Worker; CI, Confidence Interval; IG, Intervention Group; LPPKN, Lembaga Penduduk dan Pembangunan Keluarga Negara (translation: National Population and Family Development Board Malaysia); mHealth, mobile Health; OR, Odds Ratio; SEACO, South East Asian Community Observatory; WHO, World Health Organization.

## Results

We recruited 483 women in total; 122/225 from the IG and 144/258 from the CG completed the baseline and follow-up survey. Uptake of CBE by the IG was 45.8% (103/225) whilst 3.5% (5/144) of women from the CG who completed the follow-up survey reported that they attended a CBE during the study period (adjusted OR 37.21, 95% CI 14.13; 98.00, p<0.001). All IG women with a positive CBE attended a follow-up mammogram (11/11). Attendance by IG women was lower among women with a household income ≥RM 4,850 (adjusted OR 0.48, 95% CI 0.20; 0.95, p = 0.038) compared to participants with a household income <RM 4,850.

## Conclusion

The results suggested that the bespoke multicomponent mHealth intervention may be used to address the significant public health problem of low uptake of BC screening in rural Malaysia.

## Introduction

Prior to the COVID-19 pandemic, it was estimated that 1 in 10 males and 1 in 9 females will develop cancer before 75 years old and that about one-third to one-half of cancer-related deaths could be avoided through early presentation, detection and appropriate treatment. Breast cancer (BC) is the most common cancer in Malaysia with an age-standardized incidence rate (ASR) of 34.1/100,000 (2012–2016) and an age-standardised mortality rate of 12.0/100,000 (2020) [1,2]. Approximately half of BC patients tend to be diagnosed at advanced stages (48%) [2] with a considerable variation in BC incidence and prognosis across ethnic groups and geographical areas in Malaysia. For example, although BC incidence was lowest among ethnic Malay women [2], late presentation and poorer survival was more prevalent compared to other ethnic groups [3]. Women from rural areas also presented at later stages—the majority (60%) of BC patients from rural areas of Segamat District presented at stages III and IV [4].

According to the World Health Organization (WHO), early detection, treatment and management of BC are the three pillars of BC control [5,6]. Health promotion and early detection are key particularly in LMICs where late-stage detection of BC is high [6]. The most common screening modalities are clinical breast examination (CBE) and mammogram. CBE is less cost-intensive than mammograms and evidence suggests that CBEs may be as effective as a mammograms in terms of mortality [7]. In Malaysia, opportunistic CBE and mammography are the two main methods of BC screening [8]. Consequently, BC screening uptake depends on doctors offering CBE and mammogram when women at average risk of BC attend local clinics and women being aware of BC and the importance of screening and early detection.

CBE uptake in Malaysia varies between the main ethnic groups (Malay, Chinese, Indian and indigenous women) and ranges from 37% among indigenous women to 66% among Chinese Malaysians [9]. A recent survey of 993 women in Selangor State showed that only 29.7% of study participants aged ≥50 years attended BC screening [10]. A quarter of participants did not know 'how to go about BC screening', understanding about mammograms was lacking and willingness to participate in BC screening was low [10]. Negative beliefs and attitudes towards cancer and screening, lack of time and confidence to visit a doctor were also common barriers to help-seeking [10,11].

Lack of awareness about cancer signs is one of the key risk factors for patient-related delay in receiving a diagnosis [12,13], and difficulty in accessing a doctor is associated with anticipated delayed help-seeking for breast changes [13,14]. These factors are likely to help explain why BC patients in poor rural communities in Malaysia tend to be diagnosed at advanced symptom presentation [4]. Thus, providing easy-to-understand education and addressing access barriers has considerable potential to improve cancer screening uptake. Often, community health workers (CHWs) are trained to engage with communities in low- and middle-income countries (LMICs) where resources are limited and commonly they play a role in educating and assisting individuals to access health services [15,16]. For example, a recent review highlighted the role that CHWs play in promoting cervical cancer screening [15]; and, in the response to the COVID-19 pandemic, CHWs have delivered testing and vaccination programmes in collaboration with local authorities, particularly in LMICs with less resourced health care systems [17].

The COVID-19 pandemic and the resulting movement control orders (MCOs) in Malaysia, which enforced travel restrictions to limit the spread of the virus, added further barriers to accessing health services, particularly amongst rural communities, and contribute potentially to an increase in BC cases and their late detection. Globally, the pandemic has led to an increase in the use of mobile health (mHealth) and telehealth for the management and the delivery of healthcare [18–20]. The WHO supports the adoption of a 'global strategy on digital health' [21] to contribute to the advancement of the sustainable development goals [22]. Digital health is also likely to aid in the task of navigating at-risk individuals to use cancer screening services [23,24]. The main aim of this study, mindful of the context of the pandemic, and cognizant of the insights from the literature that is reviewed above, was to design, implement and evaluate an intervention to improve uptake of CBE screening in Malaysia and BC symptom recognition.

## Methods

In keeping with the UK MRC Framework for the Development and Evaluation of Complex Interventions [25], we conducted a series of research activities in order to inform and prepare a protocol for the randomised controlled trial (RCT) evaluation of the intervention [26]. This research protocol was registered with the ISRCTN registry (ISRCTN15196866; 10/08/2022) and has been reported according to the CONSORT recommendations for randomized trials [27]–see Fig 1. The research team registered the study retrospectively to meet the journal requirements. The authors confirm that all ongoing and related trials for this intervention are registered. A formal pilot study was not conducted. However, the research team tested the data collection forms internally, which did not lead to any amendments. Ethics approval has been granted by the Monash University Human Research Ethics Committee (ID: 29682).

### Setting

Malaysia is a multi-ethnic country with a population of 32.7 million in 2020 [28]. The main ethnic groups are Malays (69.6%), Chinese (22.6%) and Indians (6.8%) [28] and the majority of the population (77%) live in urban areas and 23% live in rural areas [29]. The South East Asian Community Observatory (SEACO) is a health and demographic surveillance system (HDSS) established in 2011 by Monash University and is located in Segamat, a district in the state of Johor, Peninsular Malaysia. SEACO has a comprehensive database of over 44,000 people living in Segamat, Malaysia, which is 85% of the population in 5/11 sub-districts (Sungai Segamat, Jabi, Gemereh, Bekok, and Chaah) and 24% of the total population in Segamat. It captures detailed longitudinal information about the health of Segamat residents; and provides

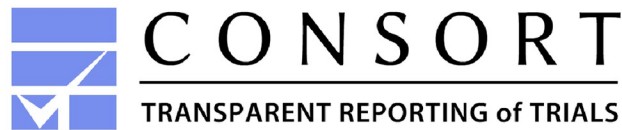

Randomized (n=1769)

**Allocation**

Allocated to intervention group (n=888)

Allocated to control group (n=881)

Excluded (n=1286)
- Not meeting inclusion criteria (n=100)
- Declined to participate (n=421)
- Uncontactable (n=758)
- Other (n=7)

**Enrollment**

Enrolled to intervention group (n=225) and received the intervention

Enrolled to intervention group (n=258)

**Follow-Up**

Lost to follow-up (give reasons) (n=103)
- Declined to participate at follow-up (n=50)
- Uncontactable (n=53)

Lost to follow-up (give reasons) (n=114)
- Declined to participate at follow-up (n=57)
- Uncontactable (n=57)

**Analysis**

Analysed (n=122)
- Excluded from analysis (n=0)

Analysed (n=144)
- Excluded from analysis (n=0)

**Fig 1. CONSORT flow diagram.**

a research platform for a focused study of screening implementation [30]. Participants were recruited from two sub-districts, i.e. Sungai Segamat–a town area, and Jabi–a rural area, which contain a mix of ethnic groups, and are representative of the population in Segamat.

## Study population

The target population comprised residents from Sungai Segamat and Jabi who completed the health survey in 2018, were recorded in the SEACO database and, previously, gave consent to be contacted about participating in research studies. Thus, we contacted women aged 40–74 years who had registered their phone number with SEACO. Only women who had a mobile phone number via which they could be contacted were able to participate in the study. BC patients and survivors, and women who reported to the research team at the time of the interview that they were experiencing BC symptoms were excluded from the research and encouraged to seek help from their local doctor as soon as possible.

## Participant randomisation and recruitment

Women from the two sub-districts in the SEACO database were randomized to the intervention group (IG) or comparator group (CG) with a 1:1 allocation prior to being contacted. Randomisation was conducted by a statistician using Stata module RANDOMIZE without blocks [31]. Participant enrolment was conducted by trained SEACO data collectors. All women who were randomised to the IG or CG were invited to participate by trained data collectors (DCs) over the phone. The resources for this study did not permit separation of enrollment and data collection procedures and, so, DCs were aware of the group to which women were assigned and this aspect may have increased the risk of bias. Participants and DCs were not blinded during the enrollment and surveys. CHWs did not collect outcome measures. Women who agreed to participate in the study received an explanatory statement about the study and a token of appreciation for their participation.

## Intervention

Our evidence synthesis [32] presented a clear narrative with respect to the need to educate rural women about BC screening and the importance of early detection of cancer combined with community navigation to encourage women to attend BC screening. IG and CG participants were interviewed at baseline and at follow-up four weeks after intervention delivery had been completed.

Women who were randomized to the IG received an intervention consisting of mHealth education and community navigation. Our previous mass media campaign, the *Be Cancer Alert Campaign* (BCAC), improved BC symptom awareness in Malaysia [10] and, so, we used the BCAC website, video and information brochure to educate women as part of the intervention for this study. The results from our research [33] indicated that women faced emotional, practical, sociocultural and health system-related barriers, and most recently, concerns regarding the COVID-19 pandemic. Therefore, we recruited and trained local female community health workers (CHWs) to allay and assuage such fears and concerns and to help women navigate barriers to the breast health care system [34]. Due to the COVID-19 pandemic and the MCO in Malaysia, the uptake intervention was delivered through mHealth. In 2021, 99.6% of Malaysian households had access to a mobile phone [35], and we therefore, decided to communicate with women through mobile phones. A multi-component mHealth intervention was used to promote screening and included a combination of telephone calls and text messages to communicate with, and navigate, participants. Additionally, we trained two male CHWs to engage with a woman's husband or male relatives if a woman communicated to her female

CHW that she would benefit from such engagement. Access to, and use of, the usual local MoH health clinics was restricted due to the pandemic. Instead, we referred women participants to a family planning clinic in Segamat that was funded by the Malaysian Government Ministry of Women, Family and Community Development (LPPKN)–the LPPKN provided free CBE screening. A one-off reimbursement for transportation fees (RM 15 / 3 USD) was offered to participants, which covered the cost of driving to and from the clinic.

**Intervention group.** Fig 2 demonstrates the CENP intervention flow. Trained data collectors (DCs) from SEACO contacted (via phone) women who were assigned to the IG and invited them to participate in the study. A baseline survey interview, that followed a systematically developed and standardized calling protocol, was conducted with women who provided verbal consent to participate in the study. Verbal consent was documented in a spreadsheet by the DCs. Next, DCs sent a text message that contained the link to the BCAC materials to women from the IG group [36]. One CHW was assigned to support between 50–60 women who agreed to participate. CHWs called the women to whom they had been assigned during the same week that the baseline assessment interview had been completed to discuss BC symptoms and BSE, address barriers to screening, and ask them if they were interested in attending a CBE at the LPPKN clinic. CBE appointments were arranged to occur the following week at a time that suited interested participants. CHWs discussed with participants who did not want to avail of the offer of screening their concerns and fears, but respected the decision of women who reiterated that they did not want to attend the CBE. Women with normal CBE results were asked by LPPKN nurses to attend screening biannually, either at the LPPKN or at a health clinic, as recommended in the clinical practice guidelines [37]. A doctor at the LPPKN clinic met with women who received an abnormal CBE finding and referred them for a mammogram at hospital as soon as possible after their CBE, free of charge. Participants were informed about their mammogram result by an LPPKN nurse. CHWs were not informed about results unless a woman chose to share the outcome of their screening tests with their CHW. CHWs called women who requested a CBE but who did not attend their scheduled CBE appointment in order to explore reasons for non-participation, address further barriers and reschedule another appointment if a participant agreed. A LPPKN nurse called women with a CBE positive result who missed their mammogram appointment to enquire if they wanted to reschedule their appointment.

**Comparator group.** The SEACO-trained DCs contacted (via phone) women who were randomised to the CG—during this call, they informed women who agreed to participate that BC is the commonest cancer amongst women in Malaysia and about the importance of early detection. Women in the CG did not receive the intervention (described above) but they could avail of 'usual' screening via their local clinic–however, elective procedures were stopped at the local government clinics to facilitate COVID-19 vaccinations and patients. In addition, CG participants were offered a scheduled free CBE at the LPPKN clinic after the follow-up data collection phase had been completed– 26/258 women availed of the offer and received a CBE at the LPPKN clinic.

## Evaluation

**Outcome measurements.** Our primary outcome measure was the proportion of women who took up CBE screening. In addition to engagement with the screening programme, secondary outcomes included change related to self-reported intention to attend CBE and mammogram screening. Outcomes related to the programme evaluation were informed by the RE-AIM framework, i.e. reach, effectiveness, adoption, implementation and maintenance which have been described in more detail previously [38], together with measures of acceptability, appropriateness and feasibility [39].

**Train 15-20 female CHWs**
Each CHW was trained and assigned to support between 50-60 women.
CHW were in close contact with clinics to schedule appointments and with research team to address any questions.

**Train 2 male CHWs**
Male CHWs were trained to provide information for women's husbands/ partners /other male relatives who the women rely on for transport and support.

**CG: Telephone recruitment & baseline survey** (n=888)
SEACO staff assessed and recruited randomly selected women (provided study information, assessed eligibility, gained verbal consent, completed baseline survey)

**IG: Telephone recruitment & baseline survey** (n=881)
SEACO staff assessed and recruit women randomised to the intervention group (provided study information, assessed eligibility,gained verbal consent, completed baseline survey)

**IG: Text message**
SEACO sent a text message with a link of the BCAC website including information on BC & BC screening (i.e. video and brochure about signs, symptoms, BSE and CBE)

**IG: Phone call from CHW**
CHW called participants after recruitment. They made sure everyone received the text message and address any further questions/ barriers and **scheduled a CBE** if participants were willing to attend screening. If 'lack of support from husbands' was a barrier for women to attend CBE, CHW offered women for male CHW to talk to husband/ other male family members.

**IG: CBE screening at LPPKN**

**CBE normal:** recommendation to screen biannually

**CBE abnormal:** referral for mammogram

**CBE refused/not attended:** CHW followed-up with women who missed their appointment to identify reasons and rescheduled if women agreed (up to 1 time)

**IG & CG: Telephone follow-up survey**
SEACO staff conducted the follow up assessment

**Fig 2. CENP study flow.**

**Methods of assessment.** *Participant information.* Information about gender, age, ethnicity, household income, marital status, education, occupation and study sub-district were extracted from the most recent health survey (2018) recorded in the SEACO database in order to present a profile of study participants. Participants were asked questions about mobile phone ownership/usage and internet usage during the baseline survey.

*Surveys (baseline and follow-up).* The baseline and follow-up survey interviews were completed over the phone. Trained SEACO DCs conducted telephone interviews with participants from the IG and CG that took approximately 15–20 minutes. The follow-up survey took between 20–25 min for the IG and 15–20 min for the CG. The survey interview comprised a number of previously adapted and validated questionnaires. CBE and mammogram screening intentions at baseline were measured by asking participants to respond to the statement, 'I intend to have a Clinical Breast Examination to check for breast cancer in the near future', using a 5-point Likert scale (strongly disagree–strongly agree), and, alongside the 5-point scale, women had the option of responding alternatively, 'I have not yet thought about this'. Next, participants were asked at what age they were thinking of availing of a CBE (unless participants chose the last or alternative option) [40]. The same questions were posed to participants regarding their intention to attend mammogram screening. CG participants were also asked 'Have you ever had your breast examined by a doctor or nurse? (i.e. Clinical Breast Examination)', and 'If yes, when was the last time you had a CBE?' to assess self-reported CBE attendance. The follow-up survey also included specific questions to assess participant satisfaction with screening [41,42] as well as their acceptability and the appropriateness of the intervention based on an adapted version of the validated Acceptability E-scale [43] (IG only).

*CBE screening attendance.* LPPKN nurses were asked to record CBE attendance for participants from the IG in a spreadsheet that was shared with SEACO on a weekly basis. CBE screening attendance was self-reported by participants from the CG and was a binary outcome measure for both groups (i.e. did OR did not attend screening).

## Statistical analysis

Quantitative data was analysed with SPSS vs 24. Descriptive statistics at baseline were reported as mean (SD) for continuous data and frequencies (percentages) for categorical data. Differences in screening uptake between the IG and CG was assessed using regression analysis (adjusting for ethnicity). A Chi-square test compared the proportion of IG and CG participants who attended CBE after the intervention had been delivered, approximately 4–6 weeks after completing the survey. Unadjusted and adjusted odds ratios were calculated using logistic regression; and linear regression was used to generate unadjusted and adjusted mean differences with 95% CIs for the screening intention score. Further regression analyses investigated the influence of factors (e.g. ethnicity, age, income, marital status, education, monthly family income and study area) in terms of affecting screening participation.

## Sampling size and procedure

A sample size of 1140 would allow approximately 90% power to detect, as statistically significant at the 5% level, an increase of 9% or more in the proportion aware of a BC symptom in the IG compared to the CG after the intervention (based upon our a priori estimates of baseline awareness of a breast lump as a cancer symptom of 65% from our previous study). A sample size of 1140 would also afford over 80% power to detect an absolute increase of 9% in the proportion of women who availed of a CBE following receipt of the intervention compared to the comparator group.

## Results

### Socio-demographic characteristics

The intervention (including baseline and follow-up calls) took place between September 2021 and January 2022. Fig 1, the CONSORT flow chart, indicates the number of women who were enrolled, allocated, followed-up and who provided data for analysis. We randomized women from Sungai Segamat and Jabi to either IG (n = 888) or CG (n = 881) prior to recruitment to account for attrition and for participants who were not interested in participating. We recruited 483 eligible women to the study—225 eligible and contactable women from the IG and 258 women from the CG completed the baseline survey interview; 122/225 (54.2%) women from the IG completed the follow-up survey interview compared to 144/258 (55.8%) women from the CG. Table 1 describes socio-demographic differences between IG and CG participants at baseline and at follow-up. At baseline, women in the IG were 67.6% Malay and 31.1% Chinese Malaysians compared to the CG with 74.4% Malays and 22.1% Chinese

**Table 1. Socio-demographic characteristics of participants at follow-up.**

|  | IG (n = 122) | % | CG (n = 144) | % |
|---|---|---|---|---|
| **Age group** |  |  |  |  |
| 40–49 | 48 | 39.3 | 59 | 41.0 |
| 50–59 | 32 | 26.2 | 40 | 27.8 |
| 60–69 | 41 | 33.6 | 40 | 27.8 |
| 70–74 | 1 | 0.8 | 5 | 3.5 |
| **Ethnic group** |  |  |  |  |
| Malay | 67 | 54.9 | 109 | 75.7 |
| Chinese | 52 | 42.6 | 33 | 22.9 |
| Other | 3 | 2.5 | 2 | 1.4 |
| **Education level** |  |  |  |  |
| No formal education | 1 | 0.8 | 2 | 1.4 |
| Primary | 29 | 23.8 | 32 | 22.2 |
| Secondary | 77 | 63.1 | 93 | 64.6 |
| Tertiary | 14 | 11.5 | 15 | 10.4 |
| **Employment**[a] |  |  |  |  |
| Not working | 78 | 63.9 | 83 | 57.6 |
| Working | 44 | 36.1 | 61 | 42.4 |
| **Household income** |  |  |  |  |
| <RM 4,850 | 99 | 81.1 | 121 | 84.0 |
| ≥RM 4,850 | 23 | 18.9 | 23 | 16.0 |
| **Sub-district** |  |  |  |  |
| Sg Segamat | 87 | 71.3 | 99 | 68.8 |
| Jabi | 35 | 28.7 | 45 | 31.3 |
| **Marital status** |  |  |  |  |
| Single[b] | 16 | 13.1 | 23 | 16.0 |
| Married | 106 | 86.9 | 121 | 84.0 |

Missing variables: Education level (n = 8).

[a] Employment in the last 30 days.

[b] Single includes those who were never married, separated, divorced and widowed.

Malaysians (p = 0.034). This pattern was similar at follow-up (IG 54.9% Malays, 42.6% Chinese Malaysians and CG 75.5% Malays and 22.9% Chinese Malaysians, p = 0.002). There were no other statistically significant differences between the two groups at baseline and follow-up.

## CBE and mammogram uptake

Data from the LPPKN clinic indicated that 103/225 (45.8%) women from the IG received a CBE. Clinic data about CBE use by the CG was not available to the research team. However, according to self-reported CBE data, 5/144 (3.5%) women from the CG compared to 71/122 (58.2%) women from the IG indicated at the post-intervention assessment point that they availed of a CBE (adjusted OR 37.21, 95% CI 14.13; 98.00, p<0.001) (Table 2). All participants (11/11) with a positive CBE attended a mammogram and one participant was diagnosed with breast cancer.

Participants with a household income ≥RM 4,850 were significantly less likely to avail of the free CBE (adjusted OR 0.48, 95% CI 0.20; 0.95, p = 0.038) compared to participants with a household income <RM 4,850 (Table 3). The odds of uptake and attending a CBE seemed to be higher amongst participants who were working/employed compared to participants who were not working albeit not statistically significant (adjusted OR 1.73, 95% CI 0.92, 3.25, p = 0.09). Furthermore, LPPKN data suggested that 10% of the CG participants took up the offer of a CBE when they were offered a free CBE *after* the study had ended.

## Intention to attend CBE screening

33/122 (27%) women from the IG reported at follow-up a positive change from baseline in terms of their intention to avail of a CBE compared to 16/144 (11%) from the CG ($\chi^2$ = 11.478, p = 0.003); 18/33 women in the IG followed through on their reported intentions and underwent a CBE at the LPPKN clinic. Similarly, 26/122 (21.3%) women from the IG and 16/144 (11%) women from the CG who reported at baseline that they had no intention to receive mammogram screening reported at follow-up that they intended to avail of a mammogram ($\chi^2$ = 5.224, p = 0.073). IG mean score at follow-up was higher (4.80, SE 0.16) compared to CG mean score (4.51, SE 0.14) indicating a significant adjusted mean difference score (0.44, 95% CI 0.06–0.83). Similarly, the mean score regarding the intention to have a mammogram was higher in the IG at follow up (4.69, SE 0.17 vs 4.42, SE 0.14) though the adjusted mean difference was not statistically significant (0.38, 95% CI -0.01–0.76) (Table 4).

## Intervention acceptability and appropriateness

About half of the participants from the IG reported that they received the link to the *Be Cancer Alert* website (67/122, 54.9%), 35/67 (52.2%) women spent time looking at the information on the website and 30/35 (85.7%) found the information on the website helpful (Table 5). The majority of the 67 women who reported that they had received the link (39/67; 58% vs 28/67;

**Table 2. Reported CBE screening uptake at follow-up.**

| | CG (n = 144) | IG (n = 122) | OR (95% CI) (unadj.) | *P* | OR (95% CI) (adj.)[b] | *P*[b] |
|---|---|---|---|---|---|---|
| **Reported CBE uptake[a]** | | | | | | |
| No | 139 (96.5) | 51 (41.8) | *1.00 (Reference)* | | *1.00 (Reference)* | |
| Yes | 5 (3.5) | 71 (58.2) | 38.70 (14.79; 101.28) | <0.001 | 37.21 (14.13; 98.00) | <0.001 |

[a] Based on self-report by CG participants and clinician-reports for IG participants.

[b] Adjusted for ethnicity.

**Table 3. Relationship between socio-demographic characteristics of IG participants and CBE screening attendance (reported by LPPKN clinics).**

| | n (%)[##] | P[a] | OR (95% CI) (unadj.) | P | OR (95% CI) (adj.)[b] | P[b] |
|---|---|---|---|---|---|---|
| **Age group** | | | | | | |
| 40–49 | 39/82 (47.6) | 0.744 | 1.00 (Reference) | 0.736 | 1.00 (Reference) | 0.837 |
| 50–59 | 28/58 (48.3) | | 1.03 (0.53; 2.02) | | 1.23 (0.60; 2.51) | |
| 60–69 | 34/78 (43.6) | | 0.85 (0.46; 1.59) | | 1.10 (0.51; 2.38) | |
| 70–74 | 2/7 (28.6) | | 0.44 (0.08; 2.41) | | 0.58 (0.09; 3.61) | |
| **Ethnicity** | | | | | | |
| Malay | 65/152 (42.8) | 0.371 | 1.00 (Reference) | 0.371 | 1.00 (Reference) | 0.623 |
| Chinese | 36/70 (51.4) | | 1.42 (0.80; 2.50) | | 1.24 (0.63; 2.43) | |
| Others | 2/3 (66.7) | | 2.68 (0.24; 30.16) | | 2.81 (0.24; 33.22) | |
| **Education level** | | | | | | |
| No formal education | 1/3 (33.3) | 0.372 | 1.00 (Reference) | 0.371 | 1.00 (Reference) | 0.370 |
| Primary | 22/55 (40.0) | | 1.33 (0.11; 15.61) | | 1.84 (0.15; 23.23) | |
| Secondary | 66/144 (45.8) | | 1.69 (0.15; 19.08) | | 2.08 (0.17; 26.24) | |
| Tertiary | 13/21 (61.9) | | 3.25 (0.25; 41.91) | | 4.64 (0.32; 67.40) | |
| **Employment[c]** | | | | | | |
| Not working | 64/154 (41.6) | 0.061 | 1.00 (Reference) | 0.062 | 1.00 (Reference) | 0.090 |
| Working | 39/71 (54.9) | | 1.71 (0.97; 3.02) | | 1.73 (0.92; 3.25) | |
| **Household income** | | | | | | |
| <RM 4,850 | 89/186 (47.8) | 0.173 | 1.00 (Reference) | 0.176 | 1.00 (Reference) | 0.038 |
| ≥RM 4,850 | 14/39 (35.9) | | 0.61 (0.30; 1.25) | | 0.43 (0.20; 0.95) | |
| **Study sub-district** | | | | | | |
| Sungai Segamat | 72/150 (48.0) | 0.344 | 1.00 (Reference) | 0.345 | 1.00 (Reference) | 0.829 |
| Jabi | 31/75 (41.3) | | 0.76 (0.44; 1.34) | | 0.93 (0.46; 1.86) | |
| **Marital status** | | | | | | |
| Single[d] | 16/33 (48.5) | 0.735 | 1.00 (Reference) | 0.736 | 1.00 (Reference) | 0.629 |
| Married | 87/192 (45.3) | | 0.88 (0.42; 1.84) | | 0.82 (0.37; 1.82) | |

n- number of participants who completed the CBE divided by the total number of participants who were offered the CBE (d–denominator).

[a] Results from the Chi-square test.

[b] Adjusted for age, ethnicity, education, working status, monthly household income, study sub-district, marital status.

[c] Employment in the last 30 days.

[d] Single includes those who were never married, separated, divorced and widowed.

**Table 4. Intention to attend CBE and mammogram screening at baseline and at follow-up.**

| | CG (mean, SE) (n = 144) | IG (mean, SE) (n = 122) | Mean difference (95% CI) (unadj.) | P | Mean difference (95% CI) (adj.)[a] | P[a] |
|---|---|---|---|---|---|---|
| CBE screening intentions (baseline) | 4.38 (0.16) | 3.93 (0.19) | -0.44 (-0.04 to 0.92) | 0.072 | N/A | |
| CBE screening intentions (follow-up) | 4.51 (0.14) | 4.80 (0.16) | 0.29 (-0.13 to 0.71) | 0.177 | 0.44 (0.06; 0.83) | 0.025 |
| Mammogram screening intentions (baseline) | 4.24 (0.16) | 4.04 (0.18) | -0.20 (-0.28 to 0.68) | 0.407 | N/A | |
| Mammogram screening intentions (follow-up) | 4.42 (0.14) | 4.69 (0.17) | 0.27 (-0.16 to 0.70) | 0.211 | 0.38 (-0.01; 0.76) | 0.055 |

[a] Adjusted for baseline intention score and ethnicity.

**Table 5. Study feedback from women in the IG at follow-up (n = 122).**

| | n (%) |
|---|---|
| Did you receive the link to the Be Cancer Alert website? | 67/122 (54.9) |
| Did you spend some time looking at the information related to breast cancer on the Be Cancer Alert Website? | 35/67 (52.2) |
| If yes, on a scale of 1–5, how easy was it for you to understand the information on the Be Cancer Alert Website? *(1 –very difficult, 5 –very easy)* | 24/35 (68.6)<br>(very easy or somewhat easy) |
| If yes, on a scale of 1 to 5, how helpful was the information on the Be Cancer Alert Website?<br>*(1 –very helpful, 5 –not helpful at all)* | 30/35 (85.7)<br>(very helpful or somewhat helpful) |
| In the last few weeks, did you talk to a CHW about breast cancer over the phone? | 94/122 (77.0) |
| ***Please rate the next five statements on a scale between 1 to 5 (1 –not at all, 5 –to a very large extend)*** | |
| You felt comfortable talking to the CHW about breast cancer and breast cancer screening. | 87/94 (92.6)<br>(to a very large extend or to a large extend) |
| The CHW took your concerns seriously. | 86/94 (91.5)<br>(to a very large extend or to a large extend) |
| The CHW provided enough time for dialogue | 91/94 (96.8)<br>(to a very large extend or to a large extend) |
| The CHW was easy to understand. | 89/94 (94.7)<br>(to a very large extend or to a large extend) |
| The CHW was competent. | 91/94 (96.8)<br>(to a very large extend or to a large extend) |
| ***Please rate the next five statements on a scale between 1 to 5 (1 –not at all, 5 –to a very large extend)*** | |
| What challenges did you face with attending the Clinical Breast Examination?* *(Tick all that apply)* | It was easy for me and I did not have any challenges. (n = 55)<br>Lack of transport (n = 5)<br>Fear of screening (n = 3)<br>Fear of cancer (n = 1)<br>Embarassement (n = 1)<br>Lack of time to go to the CBE (n = 6)<br>Others (n = 7) |
| The nurse took your concerns seriously.* | 55/71 (77.5)<br>(to a very large or to a large extend) |
| The nurse cared for you.* | 59/71 (83.1)<br>(to a very large or to a large extend) |
| The nurse provided enough time for dialogue.* | 57/71 (80.3)<br>(to a very large or to a large extend) |
| The nurse was easy to understand.* | 52/71 (73.2)<br>(to a very large or to a large extend) |
| The nurse was competent* | 58/71 (81.7)<br>(to a very large or to a large extend) |
| ***The last few questions are about your experience with the breast cancer screening programme.*** | |
| On a scale of 1–4, how satisfied were you with the screening services?* *(1-very dissatisfied, 4-very satisfied)* | 59/71 (83.1)<br>(very satisfied/ satisfied) |
| On a scale of 1–4, how likely are you to have the breast cancer screening (CBE) done again by this institution?* *(1-very unlikely, 4-very likely)* | 60/71 (84.5)<br>(very likely or somewhat likely) |

*(Continued)*

**Table 5.** (Continued)

| | n (%) |
|---|---|
| On a scale of 1–4, how likely are you to recommend the screening services to others?* *(1-very unlikely, 4-very likely)* | 59/71 (83.1) (very likely or somewhat likely) |
| On a scale of 1–4, how much did you trust the screening results?* *(1-did not trust them at all, 4 –trusted them completely)* | 59/71 (83.1) (strongly trusted them or somewhat trusted them) |
| What would be your preferred method of contact if your clinic wanted to invite you to participate in cancer screening again? | Phone call: 98/122 (80.3) Text message: 39/122 (32.0) Letter: 25/122 (20.5) Face-to-face invitation from doctor/nurse: 10/122 (8.2) I don't want to be invite for screening again: 6/122 (4.9) No reply: 2/122 (1.6) |

*These responses are only from women who completed the CBE at the LPPKN clinic.

42%) attended the LPPKN for a CBE. More than two-fifths of women stated that they did not receive the link (45) or could not recall if they had received the link (2); and 24/45 (53%) attended the LPPKN clinic for a CBE. Over 90% of women who were contacted by a CHW reported that they felt comfortable speaking to the CHW, felt taken seriously, were given enough time for dialogue and that the CHW was easy to understand and competent. Most women (80%) who received a CBE at the LPPKN clinic agreed to a large/very large extent that the nurses at the clinic took their concerns seriously, cared for them, provided enough time for dialogue and were competent.

### Participants and women who declined to participate

Table 6 suggests that contactable participants were significantly different to non-participants in terms of their age, ethnicity, education level and study sub-district. The majority of participants were Malay (71.2%), followed by Chinese Malaysians (26.3%), they had completed secondary education (63.8%), reported an average household income of <RM 4,850 (1,024 USD), were married (83.6%) and lived in Sungai Segamat (66.9%). The number of Indian Malaysians, the indigenous community and non-citizens who were contactable and met the study inclusion criteria was very small and we have reported this small number, collectively, as 'others' from here on (n = 12 at baseline). Differences were observed between ethnic groups at follow-up (Table 7).

### Discussion

This was the first randomized study in Malaysia to test the effectiveness of a bespoke multi-component intervention to address the significant public health problem of low uptake of breast cancer screening. Overall, the results suggested that the intervention achieved positive outcomes in terms of improved CBE uptake and an increase in the number of women who reported intention to screen. CBE uptake was significantly higher in the IG (46%) compared to the CG (4%) despite the pandemic and the resulting restrictions in movement across sub-districts. The same option (but without the uptake intervention) of free CBE screening at the LPPKN clinic was offered to CG women immediately *after* study completion and 10% availed of the offer (compared to 46% of IG women).

**Table 6. Profile of participants vs non-participants at baseline.**

| | Agreed (n = 483) | % | Declined (n = 421) | % | p-value |
|---|---|---|---|---|---|
| **Age group** | | | | | |
| 40–49 | 186 | 38.5 | 130 | 30.9 | 0.002 |
| 50–59 | 130 | 26.9 | 100 | 23.8 | |
| 60–69 | 152 | 31.5 | 161 | 38.2 | |
| 70–74 | 15 | 3.1 | 30 | 7.1 | |
| **Ethnic group** | | | | | |
| Malay | 344 | 71.2 | 345 | 81.9 | <0.001 |
| Chinese | 127 | 26.3 | 72 | 17.1 | |
| Other | 12 | 2.5 | 4 | 1.0 | |
| **Education level** | | | | | |
| No formal education | 9 | 1.9 | 11 | 2.6 | <0.001 |
| Primary | 114 | 23.6 | 167 | 39.7 | |
| Secondary | 308 | 63.8 | 202 | 48.0 | |
| Tertiary | 47 | 9.7 | 36 | 8.6 | |
| **Employment** (in the last 30 days) | | | | | |
| Not working | 309 | 64.0 | 284 | 67.5 | 0.271 |
| Working | 174 | 36.0 | 137 | 32.5 | |
| **Household income** | | | | | |
| <RM 4,850 | 397 | 82.2 | 352 | 83.6 | 0.573 |
| ≥RM 4,850 | 86 | 17.8 | 69 | 16.4 | |
| **Sub-district** | | | | | |
| Sg Segamat | 323 | 66.9 | 231 | 54.9 | <0.001 |
| Jabi | 160 | 33.1 | 190 | 45.1 | |
| **Marital status** | | | | | |
| Single | 79 | 16.4 | 87 | 20.7 | 0.095 |
| Married | 404 | 83.6 | 334 | 79.3 | |

The components of the intervention or programme were informed by our earlier studies [32,33,44] and comprised community education delivered via mHealth and the use of trained CHWs to navigate women to undertake screening. The outcomes for women who received the intervention were compared to women who could avail of usual service arrangements i.e. opportunistic BC screening practice. It is important to note that the delivery and testing of the intervention occurred during the COVID-19 pandemic and unlike usual service arrangements, women were navigated to undergo screening at a LPPKN clinic since public health clinics (KKs) prioritised COVID-19 patients and were very limited in their capacity to conduct opportunistic screening. A decline in access to and use of health services during the pandemic was observed in health systems across the world, including Malaysia [45]. The decline in use of the range of LPPKN services due to the MCO meant that the LPPKN clinic was in a position to increase capacity for screening appointments. The LPPKN clinics, which were initially closed at the beginning of the MCO, were reopened when public health advocates complained about the disruptions to sexual and reproductive healthcare access during the pandemic. Consequently, the LPPKN clinics were able to function in a more agile and responsive way (e.g. providing walk-in healthcare screening services) compared to the public health clinic in each community area (which operated on an appointment only basis and had shortened opening hours during the pandemic in Malaysia).

**Table 7. Profile of participants and non-participants at follow-up (IG and CG combined)\*.**

| | Agreed (n = 266) | % | Declined (n = 107) | % | p-value |
|---|---|---|---|---|---|
| **Age group** | | | | | |
| 40–49 | 107 | 40.2 | 40 | 37.4 | 0.859 |
| 50–59 | 72 | 27.1 | 27 | 25.2 | |
| 60–69 | 81 | 30.5 | 37 | 34.6 | |
| 70–74 | 6 | 2.3 | 3 | 2.8 | |
| **Ethnic group** | | | | | |
| Malay | 176 | 66.2 | 87 | 81.3 | 0.012 |
| Chinese | 85 | 32.0 | 18 | 16.8 | |
| Other | 5 | 1.9 | 2 | 1.9 | |
| **Education level** | | | | | |
| No formal education | 3 | 1.1 | 4 | 3.7 | 0.339 |
| Primary | 61 | 22.9 | 26 | 24.3 | |
| Secondary | 170 | 63.9 | 66 | 61.7 | |
| Tertiary | 29 | 10.9 | 9 | 8.4 | |
| **Employment** (in the last 30 days) | | | | | |
| Not working | 161 | 60.5 | 72 | 67.3 | 0.222 |
| Working | 105 | 39.5 | 35 | 32.7 | |
| **Household income** | | | | | |
| <RM 4,850 | 220 | 82.7 | 88 | 82.2 | 0.915 |
| ≥RM 4,850 | 46 | 17.3 | 19 | 17.8 | |
| **Sub-district** | | | | | |
| Sg Segamat | 186 | 69.9 | 64 | 59.8 | 0.060 |
| Jabi | 80 | 30.1 | 43 | 40.2 | |
| **Marital status** | | | | | |
| Single | 39 | 14.7 | 17 | 15.9 | 0.764 |
| Married | 227 | 85.3 | 90 | 84.1 | |

\* n = 110 participants were uncontactable at follow-up.

All IG women who received a positive CBE attended mammogram screening (11/11). Similar positive results, overall, were reported for an education and navigation intervention in Jordan where women availed of free mammogram screening though the study intervention occurred prior to the pandemic and CHW interaction was face-to-face rather than over the phone [46]. Findings from studies set in high- as well as low- and middle-income countries suggested that around 3% of women with abnormal CBEs are diagnosed with BC [47,48].

There were no significant differences between Malays and Chinese Malaysians in terms of CBE attendance (and there was an insufficient number of Indian Malaysians and members of the indigenous community in the study sample to make meaningful comparisons between these ethnic groups). However, the analysis indicated that women from low income households in the IG were more likely to attend the free screening service similar to findings from our recent colorectal cancer study where participants from the PeKa B40 category (i.e. citizens in the bottom 40% household income range of <RM 4,850) were more likely to use the free screening test [49]. It appears that the financial savings of participating in free screening have greater salience for women in low-income households (many of whom are less likely to have health insurance) [50] compared to high-income households and, therefore, are more likely to participate in a screening test free of charge to avoid associated costs, compared to higher

income earners who may have access to mammograms at private hospitals. The results suggested also that the intervention led participants to express an intention to uptake a CBE or mammogram in the near future and this change in intention was particularly salient among Chinese Malaysian women compared to Malay women. This finding is in line with previous survey findings from Segamat Malaysia where Chinese Malaysian women reported much higher mammogram uptake compared to Malays (34.4% vs 16.6%) [33].

Overall, this multicomponent way of trying to increase uptake of CBE screening appeared to be acceptable to Malay and Chinese Malaysian women though it is difficult to discern the extent to which components exerted differential impacts. There was relatively low engagement with the mHealth component which comprised online information about the signs and symptoms of BC, the benefits of BC screening as well as cancer survivor stories, whereas local CHWs were described by participants in positive terms and, for example, as delivering easy-to-understand information about BC screening. Few studies have explored the role of CHWs in terms of navigating women to BC services compared to studies that have investigated the use of CHWs to promote BC education [16,34,46]. Previous research has suggested that BC education, screening and navigation were the three top areas that benefited from CHW involvement in the US, South Africa and Bangladesh [16] and may benefit, in particular, ethnically diverse populations [51]. A clustered controlled trial in Bangladesh reported a significantly higher uptake in CBE screening amongst women who received a smartphone app for data collection and a motivational video as well as CHW navigation–women in the intervention group were more likely to attend compared to women who received the smartphone only or women who received no intervention [52]. The results of this study appear to point to CHWs as playing an influential role as navigators of women to BC screening and at low cost compared to health professionals. CHWs may benefit too in terms of their increased cancer health literacy via training; and their location in local communities, and their ready availability and low cost may contribute positively to the sustainability of uptake (and other) interventions.

The LPPKN clinic was also a significant intervention component as it demonstrated flexibility in its willingness and efforts to facilitate additional CBEs on a weekly basis (as well as engaging fully with the research team). In addition, the LPPKN clinic appeared to be an acceptable location for many women to receive their CBE (perhaps a preferred location compared to usual community clinics). A total of 58 LPPKN clinics throughout Malaysia [53] form part of a national task force for breast and cervical cancer screening. However, the services that the LPPKN clinics offer are not well known to the public. The results of this study suggest that there may be merit in considering how LPPKN clinics might be used to contribute to public health efforts to meet BC screening guidelines set by the Ministry of Health in Malaysia [54].

It is important to note that there are several study limitations. The difficulty related to obtaining clinic data about CBE uptake by the CG meant that the comparative analysis between the IG and CG was limited. We recruited about 50% of the target sample size and there was considerable attrition at the follow-up interview stage which is lower than a pre-pandemic face-to-face survey that our research team conducted as part of a previous study (68% attrition at follow-up) [10]. This response pattern may be related to the telephone mode of recruitment and the collection of data by telephone survey interview albeit necessary in the context of the pandemic and the imposition of MCOs in Malaysia. Traditional face-to-face data collection may be more suitable for building rapport and trust with participants though, overall, research suggests that, the quality of data may be comparable across interview modes [55]. The profile of women who agreed to participate in the study at baseline differed from women who declined in terms of age, ethnicity, education level and study sub-district. Indian Malaysians were underrepresented and, therefore, findings are limited to the two largest ethnic groups—Malay and Chinese Malaysian women and to subsets within those groups. The data

collected by the LPPKN is a strength of this study—most CHW navigation studies are based on self-reported data only and, in addition, few CHW-related intervention studies have used a CG [16]. The self-reported CBE attendance data from the IG vs the LPPKN clinic data (58% vs 48%) suggests that some participants might have received a CBE at a different clinic or that women were biased in their response and the self-reported figures need to be treated with that in mind.

Overall, and bearing in mind the above noted limitations, the results of this study suggest that the use of a multicomponent community education and navigation intervention or programme has considerable potential to elicit positive health protective intentions and improve clinical breast examination uptake in Segamat Malaysia even in the highly challenging circumstances of the COVID-19 pandemic. The results point to ways in which the intervention might be improved as well as the need for further research to test how well the intervention works in a non-pandemic context and how its generalisability might be addressed and extended.

## Supporting information

**S1 Checklist. CONSORT-2010-Checklist.**
(PDF)

**S1 File. Protocol Open Science Framework (OSF).**
(DOCX)

**S2 File. ISRCTN registry study record 42195.**
(PDF)

**S3 File. PLOS ONE inclusivity in global research text.**
(PDF)

## Acknowledgments

We would like to thank nurses from the LPPKN clinic in Segamat for facilitating CBE and mammograms for this study, all community health workers and study participants as well as the SEACO field team involved.

## Author Contributions

**Conceptualization:** Désirée Schliemann, Aminatul Saadiah Abdul Jamil, Devi Mohan, Min Min Tan, Nur Aishah Taib, Tin Tin Su, Michael Donnelly.

**Data curation:** Aminatul Saadiah Abdul Jamil, Roshidi Ismail.

**Formal analysis:** Désirée Schliemann, Christopher R. Cardwell, Roshidi Ismail.

**Funding acquisition:** Tin Tin Su, Michael Donnelly.

**Investigation:** Désirée Schliemann, Aminatul Saadiah Abdul Jamil, Michael Donnelly.

**Methodology:** Désirée Schliemann, Aminatul Saadiah Abdul Jamil, Devi Mohan, Min Min Tan, Christopher R. Cardwell, Roshidi Ismail, Nur Aishah Taib, Tin Tin Su, Michael Donnelly.

**Project administration:** Désirée Schliemann, Aminatul Saadiah Abdul Jamil, Tin Tin Su.

**Resources:** Tin Tin Su, Michael Donnelly.

**Supervision:** Tin Tin Su, Michael Donnelly.

**Writing – original draft:** Désirée Schliemann.

**Writing – review & editing:** Aminatul Saadiah Abdul Jamil, Devi Mohan, Min Min Tan, Christopher R. Cardwell, Roshidi Ismail, Nur Aishah Taib, Tin Tin Su, Michael Donnelly.

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
