## [Decision Letter · Decision Letter 0]

2 Oct 2022

PONE-D-22-18035The development and evaluation of a mHealth, community education and navigation intervention to improve clinical breast examination uptake in Segamat Malaysia: a randomised control trialPLOS ONE

Dear Dr. Schliemann,

Thank you for submitting your manuscript to PLOS ONE. After careful consideration, we feel that it has merit but does not fully meet PLOS ONE’s publication criteria as it currently stands. Therefore, we invite you to submit a revised version of the manuscript that addresses the points raised during the review process.

We look forward to receiving your revised manuscript.

Kind regards,

Abraham Tamirat Gizaw, PhD candidate

Academic Editor

PLOS ONE

Journal Requirements:

2. Please include a complete copy of PLOS’ questionnaire on inclusivity in global research in your revised manuscript. Our policy for research in this area aims to improve transparency in the reporting of research performed outside of researchers’ own country or community. The policy applies to researchers who have travelled to a different country to conduct research, research with Indigenous populations or their lands, and research on cultural artefacts. The questionnaire can also be requested at the journal’s discretion for any other submissions, even if these conditions are not met.  Please find more information on the policy and a link to download a blank copy of the questionnaire here: https://journals.plos.org/plosone/s/best-practices-in-research-reporting. Please upload a completed version of your questionnaire as Supporting Information when you resubmit your manuscript

3. In the ethics statement in the Methods, you have specified that verbal consent was obtained. Please provide additional details regarding how this consent was documented and witnessed, and state whether this was approved by the IRB

4. "Thank you for submitting your clinical trial to PLOS ONE and for providing the name of the registry and the registration number. The information in the registry entry suggests that your trial was registered after patient recruitment began. PLOS ONE strongly encourages authors to register all trials before recruiting the first participant in a study.

1) your reasons for your delay in registering this study (after enrolment of participants started);

2) confirmation that all related trials are registered by stating: “The authors confirm that all ongoing and related trials for this drug/intervention are registered”."

Reviewers' comments:

Reviewer's Responses to Questions

**Comments to the Author**

1. Is the manuscript technically sound, and do the data support the conclusions?

Reviewer #1: Yes

Reviewer #2: No

2. Has the statistical analysis been performed appropriately and rigorously? 

Reviewer #1: Yes

Reviewer #2: No

3. Have the authors made all data underlying the findings in their manuscript fully available?

Reviewer #1: No

Reviewer #2: Yes

4. Is the manuscript presented in an intelligible fashion and written in standard English?

Reviewer #1: Yes

Reviewer #2: Yes

5. Review Comments to the Author

Reviewer #1: General comments:

Breast cancer is a threat to women especially in Low and Middle Income Countries (LMICs) as presented by the authors. Interventions aimed at preventing and treating breast cancer are encouraged. Therefore, this manuscript is addressing an important topic. In general, the manuscript is well written but due to the small sample size did not meet the required strength to detect differences between the two groups, intervention and comparator. The authors need to recalculate the resulting detection power from the achieved sample size.

Specific comments

Introduction

1. Article 1 is over referenced in the first paragraph

2. Under line 84, I suggest using the word, “participants” instead of the, “sample”.

3. Line 90, “These factors explain why BC patients…”

4. Line 102, do you mean that travel restrictions resulted to an increase of Breast Cancer cases?

Methods

Study population

1. Line 137, only women who had a mobile phone number … were considered for participation. If it is true that mobile phone ownership is an indication of a socioeconomic status, especially in LMICs, then this criterion introduced a bias that the authors should cite as a limitation. Also, it is important to provide information about the proportion of the population that owns a mobile phone.

2. Line 178… A one-off reimbursement for transportation fees to the clinic was offered to participants. How much was the transport fees, and was it enough to influence participants’ decision?

Comparator

1. Line 208, elective procedures were stopped at the local government clinics to facilitate COVID-19 vaccinations and patients. Could this act minimize alternatives and hence increase demand for the services from the study clinic? Meaning, the demand for CBE was not only due to the intervention. Unfortunately, there are no data from the comparator group on CBE.

2. Consider providing more information, for example, the purpose for BC screening attendance (Line 249) and Qualitative interviews (Line 253).

Sample size and procedures

1. Line 270, align information about sample size with that provided in Figure 1. For example, at what stage were participants allocated to the two arms?

2. Line 277, indicates that a pilot study was conducted. If the study findings were published, then I suggest referencing the article. However, if the pilot study findings have not been published, then the authors should consider providing a summary of the pilot study as part of the study methods.

Results

1. Line 288, consider providing conversion rate of the currency to US dollar.

2. Line 306, Instead of using the term, “Single” to mean those who have never married, separated/divorced and widowed, you could use terms like, “Has a partner/No partner”

Reviewer #2: Abstract

- CBE is mentioned in the introduction of the abstract without being stated in full (as has been done with BC and mHealth)

- the methods of the abstract says 'regression analyses were conducted to investigate between-group differences over time...'; it would be better to use this opportunity to state what the outcomes were, i.e. 'between group differences in uptake of screening and attendance of CBE...'

- the results in the abstract are confusing, perhaps because they don't focus or highlight on the overall effect of the intervention on the main outcome(s) of the trial. Why do they mention household income at all? This results section should first describe the effect of the intervention on the main outcomes, and if there is anything further to say about subgroups then those can be subsequently mentioned; otherwise it may look like the authors are cherry-picking 'significant' results to report.

Introduction

- it seems a little out of place to mention the statistics on the rates of all cancers in males and females between two sentences talking about breast cancer (lines 63-65 and 65-67) - unless these lines are also referring to breast cancer, in which case this needs to be written more clearly.

- the statement in line 159-161 about the CONSORT diagram is a result and should be moved to the results section.

Methods

- in line 215, what you probably mean to say is that your primary measure was CBE screening uptake (perhaps more clearly, the proportion of women to took up CBE screening); the 'difference' is not an outcome but a measure of effect. Similarly, in line 219 your secondary outcomes are self-reported intention to attend CBE and mammogram screening - saying 'nature and degree...' is non-specific language in what should be unambiguous.

- in line 220, there is need for further clarity i.e. to state explicitly which outcomes were informed by the RE-AIM framework.

- reading lines 232-247, it is starting to seem to me that the primary outcome of CBE uptake was a categorical (perhaps ordered categorical) variable, although I can also see how a binary outcome could have been derived out of it. And the only reason I am thinking about a binary outcome is the previous wording of how this outcome has been described and also the mention of the proportions of CBE uptake in the abstract. What I am getting at here is that the actual outcome is not clearly described anywhere. If a binary outcome was derived from the likert scale, this needs to be clearly described. If the likert scale was used as is, this also needs to be clearly described (and if this is what was done then some aspects of the results don't make sense).

- if the primary outcome was binary, then although you could use chi-squared tests for a basic analysis, it would be better to use logistic regression models, if only so that you can present unadjusted and adjusted estimates of effect. In this case, chi-squared tests would be unnecessary. If the primary outcome was categorical, then this approach is fine, although you could also use multinomial regression models to obtain unadjusted and adjusted effects. If the outcome is ordered categorical, then you could use ordinal logistic regressions to obtain unadjusted and adjusted effect. Again, given the lack of clarity on exactly what the primary outcome is, I am unable to determine which of these I would recommend.

- again, for continuous outcomes, there is no need to use t-tests when you could use linear regression models to obtain unadjusted and adjusted effects. Here you don't need the change-in-screening intention score: you should simply regress the endpoint scores on treatment group (and any covariates, in the adjusted model) further adjusting for the baseline scores; this is an analysis of covariance model, which is the most statistically efficient approach for a continuous outcome with a baseline measure (and which gives identical estimates to a regression of change-from-baseline scores adjusting for baseline, given that baseline adjustment is pertinent for the improved precision of effect estimates).

- the analysis exploring factors affecting screening participation is a secondary question and should not be conflated with that seeking to estimate the effect of the intervention.

- I was not able to reproduce your sample size calculation. In my calculations, you would need about 410 participants in each arm (i.e. 820 total for two arms with 1:1 allocation) to detect a 9% absolute improvement in a binary outcome assuming a baseline level of 65%, with 80% power at the 5% level of significance for a two-tailed test. Your calculation also mentions some adjustment of recruitment to account for attrition; you would normally indicate your anticipated level of attrition as this would normally be included in the calculation

- regarding adjustments, you would normally adjust for age (and sex if relevant, but may not be relevant here if your sample was all/mostly female) and any baseline factors relevant during randomisation.

Results

- the way the results have been presented distract from the objectives of the study, in my opinion. Since the study aimed to investigate the effect of the intervention, the results should have began with a summary of the CONSORT flow diagram i.e. the numbers of participants in the treatment arms that were eligible, screened, randomised, followed-up, and for whom outcomes were available. This should have then been followed up by a table of the baseline characteristics of participants with outcomes to be included in the main analysis, split by treatment group. Crucially, this table must not include statistical tests/p-values comparing the treatment arms with respect to baseline characteristics, because in a randomised trial it is expected that the groups would be balanced and any imbalance is random and not particularly statistically interesting. Thus, the p-values in Table 3 are unnecessary, as are the separate columns for 'baseline' and 'follow-up' - this table needs to focus on the participants for whom outcomes are available and are included in the comparison of treatment groups. Tables 1 and 2 which compare characteristics of women who are included and not included are addressing a secondary question which, although interesting and even crucial in this case given the large amount of losses/attrition, does not belong to this section of the results.

- a results table showing the counts and proportions of participants with each binary outcome in each treatment group, along with unadjusted and adjusted effects (odds ratios) with 95% confidence intervals and p-values (and no other information, i.e. the main results) should be presented (e.g. Table 4 in https://www.sciencedirect.com/science/article/pii/S0140673618317823 ignoring the 'risk difference' column.)

- (notwithstanding my comment above about the relevance of the analysis presented in Table 4) the Wald test p-values in the 5th and 7th columns for age group, ethnicity and education level should be replaced with overall (e.g. likelihood ratio) p-values, as these are the appropriate statistic to evaluate whether each variable is associated with screening attendance. The currently presented Wald test p-values only tell you which level of each variable is different from the comparison category, and they change depending on the comparison category, and don't tell you anything about the overall association between each variable and screening attendance.

- The results in Table 5 are inappropriately presented (and this stems from the inappropriateness of the analytical approach - see my suggested analytical approach above). The appropriate way to present this would be to have the mean and standard error (not standard deviation) of each outcome measure by treatment arm, followed by the unadjusted and adjusted mean differences with 95% confidence intervals and p-values, for example Table 2 in https://www.sciencedirect.com/science/article/pii/S0140673618317823. The columns indicating the group means and SEs should indicate the numbers of individuals included in the analysis.

- It is not clear which objective the analysis presented in Table 6 is addressing.

- There is a lot of attrition/losses in this trial and the number of participants eventually included in the analysis is probably smaller than what was required under the original assumptions (see my comment on the sample size calculation). A clear recognition of this should be made in the manuscript, and this is the part where some analysis of how the individuals included versus not compare (i.e. Tables 1 and 2) belong, but only at the end of the results after the analyses addressing the main objectives have been presented in a standard manner.

Overall comments

This is an important study. However, analysis and reporting need to be improved. I would suggest that the authors use the CONSORT checklist to aid their reporting, that they focus the presentation of the analysis and results on the objectives of the trial, to report the results in a manner consistent with what is expected of clinical trial reports, and to clarify their sample size calculation, addressing the issues I have raised.

6. PLOS authors have the option to publish the peer review history of their article (what does this mean?). If published, this will include your full peer review and any attached files.

Reviewer #1: **Yes: **BENARD OMONDI OCHIENG

Reviewer #2: No

---

## [Author Response · Author response to Decision Letter 0]

8 Mar 2023

Dear PLOS One Editorial Team, 

Many thanks for giving us the opportunity to revise our manuscript entitled, ‘The development and evaluation of a mHealth, community education and navigation intervention to improve clinical breast examination uptake in Segamat Malaysia: a randomised control trial’. We would like to thank the Editor and Reviewers for their comments on this manuscript. We have attempted to address all comments as outlined below and highlighted in the marked copy of the manuscript. Please note that the order of the tables has changed, hopefully this is clear in the manuscript. 

If there are any further questions or suggestions for improvement, please do not hesitate to get in touch. 

Kind regards, 

Dr Desiree Schliemann

(on behalf of the study team) 

Journal Requirements:

Please ensure that your manuscript meets PLOS ONE's style requirements, including those for file naming. The PLOS ONE style templates can be found at https://journals.plos.org/plosone/s/file?id=wjVg/PLOSOne_formatting_sample_main_body.pdf and 

Reply: We followed the guidelines. 

2. Please include a complete copy of PLOS’ questionnaire on inclusivity in global research in your revised manuscript. Our policy for research in this area aims to improve transparency in the reporting of research performed outside of researchers’ own country or community. The policy applies to researchers who have travelled to a different country to conduct research, research with Indigenous populations or their lands, and research on cultural artefacts. The questionnaire can also be requested at the journal’s discretion for any other submissions, even if these conditions are not met. Please find more information on the policy and a link to download a blank copy of the questionnaire here: https://journals.plos.org/plosone/s/best-practices-in-research-reporting. Please upload a completed version of your questionnaire as Supporting Information when you resubmit your manuscript

Reply: We have completed and attached this document. 

3. In the ethics statement in the Methods, you have specified that verbal consent was obtained. Please provide additional details regarding how this consent was documented and witnessed, and state whether this was approved by the IRB

Reply: This was obtained over the phone after women were read the study information and were given time to consider their response. The interviewer/data collector followed a systematically developed and standardised calling protocol and recorded the response of each participant in a spreadsheet. This was approved by the ethics committee. We added this information to the protocol. 

4. "Thank you for submitting your clinical trial to PLOS ONE and for providing the name of the registry and the registration number. The information in the registry entry suggests that your trial was registered after patient recruitment began. PLOS ONE strongly encourages authors to register all trials before recruiting the first participant in a study.

1) your reasons for your delay in registering this study (after enrolment of participants started);

2) confirmation that all related trials are registered by stating: “The authors confirm that all ongoing and related trials for this drug/intervention are registered”."

Reply: The research team did not view the study strictly in terms of a clinical trial design (eg there is minimum available data about the comparator group). This study was not testing efficacy but effectiveness of the intervention in a real world setting which is also why we did not view this as a clinical trial. However, we registered the study following a request from the journal and in keeping with journal requirements and this timeline and order of events explains the delay in study registration. 

Reviewers' comments:

Reviewer-1 

General comments:

Breast cancer is a threat to women especially in Low and Middle Income Countries (LMICs) as presented by the authors. Interventions aimed at preventing and treating breast cancer are encouraged. Therefore, this manuscript is addressing an important topic. In general, the manuscript is well written but due to the small sample size did not meet the required strength to detect differences between the two groups, intervention and comparator. The authors need to recalculate the resulting detection power from the achieved sample size.

Reply: We have noted this study power limitation and updated the sample size paragraph, accordingly. 

Specific comments

1. Article 1 is over referenced in the first paragraph

Reply: We have reduced the number of times that article number one is referenced in the first paragraph (reference one - the current national cancer report for Malaysia - reports incidence and prevalence rates and, so, it provides support for the statistical profile that underpins the need for the study). 

2. Under line 84, I suggest using the word, “participants” instead of the, “sample”.

Reply: We have changed the wording. 

3. Line 90, “These factors explain why BC patients…”

Reply: We have amended the wording to read that, ‘These factors are likely to help explain…’ in order to indicate that the factors regarding why cancer patients are diagnosed at a late stage are under investigated in the target population and communities. 

4. Line 102, do you mean that travel restrictions resulted to an increase of Breast Cancer cases?

Reply: We think that women with breast cancer signs and symptoms may have been encouraged to postpone to visit their doctor as they were being asked to stay at home or not travel further than a certain distance and, therefore, breast cancer cases may not have been detected until a later stage than may have been the case in usual circumstances. Moreover, this impact on time of presentation is likely to be a common finding across most health care systems.

Methods

Study population

1. Line 137, only women who had a mobile phone number … were considered for participation. If it is true that mobile phone ownership is an indication of a socioeconomic status, especially in LMICs, then this criterion introduced a bias that the authors should cite as a limitation. Also, it is important to provide information about the proportion of the population that owns a mobile phone.

Reply: We have included statistics on mobile phone ownership in the intervention section. Since over 99% of households in Malaysia own a mobile phone, we did not see this as a limitation to this study. 

2. Line 178… A one-off reimbursement for transportation fees to the clinic was offered to participants. How much was the transport fees, and was it enough to influence participants’ decision?

Reply: SEACO provides a one-off MYR15 (USD3) travel reimbursement to get a CBE at the LPPKN clinic. This covered private taxi hire or petrol costs for driving to and from the clinic. We included this information in the manuscript. 

Comparator

1. Line 208, elective procedures were stopped at the local government clinics to facilitate COVID-19 vaccinations and patients. Could this act minimize alternatives and hence increase demand for the services from the study clinic? Meaning, the demand for CBE was not only due to the intervention. Unfortunately, there are no data from the comparator group on CBE.

Reply: The Discussion notes now that it is possible that the unavailability of opportunistic CBE at the local community clinics due to the pandemic may have increased demand for use of CBE at the study clinic though use of the study clinic for CBE was initiated and facilitated only via intervention community health workers who, for example, make an appointment for each woman. Women did not make appointments themselves, at least not IG participants. 

2. Consider providing more information, for example, the purpose for BC screening attendance (Line 249) and Qualitative interviews (Line 253).

Reply: Although we are a little unsure about the meaning of this comment, we have attempted to provide further information content. 

Sample size and procedures

1. Line 270, align information about sample size with that provided in Figure 1. For example, at what stage were participants allocated to the two arms?

Reply: We have updated this paragraph accordingly. 

2. Line 277, indicates that a pilot study was conducted. If the study findings were published, then I suggest referencing the article. However, if the pilot study findings have not been published, then the authors should consider providing a summary of the pilot study as part of the study methods.

Reply: We have added the following to the manuscript under methods: A formal pilot study was not conducted. However, internal informal testing of the data collection forms by the research team did not lead to any amendments.

Results

1. Line 288, consider providing conversion rate of the currency to US dollar.

Reply: We have included the conversion rate.

2. Line 306, Instead of using the term, “Single” to mean those who have never married, separated/divorced and widowed, you could use terms like, “Has a partner/No partner”

Reply: We used the term ‘single’ in this way because it appears to be common reporting practice in research papers about the topic and, so, its use facilitates easy comparisons. Also, extramarital or premarital sex is taboo in the local context, the study team did not collect this sensitive information (i.e. has a partner who they are not married to) which may put off some participants to participate in the study.

Reviewer-2

Abstract

- CBE is mentioned in the introduction of the abstract without being stated in full (as has been done with BC and mHealth)

Reply: We have spelled out CBE in the abstract. 

- the methods of the abstract says 'regression analyses were conducted to investigate between-group differences over time...'; it would be better to use this opportunity to state what the outcomes were, i.e. 'between group differences in uptake of screening and attendance of CBE...'

Reply: We have amended the abstract accordingly and we have noted in various sections of the paper that there were differences in the method of assessing and reporting outcomes (eg self-reported vs reported by the LPPKN clinic staff) and, therefore, only tentative comparisons could be made. 

- the results in the abstract are confusing, perhaps because they don't focus or highlight on the overall effect of the intervention on the main outcome(s) of the trial. Why do they mention household income at all? This results section should first describe the effect of the intervention on the main outcomes, and if there is anything further to say about subgroups then those can be subsequently mentioned; otherwise it may look like the authors are cherry-picking 'significant' results to report.

Reply: We have amended the abstract accordingly (to show uptake of CBE by IG women compared to CG women). 

Introduction

- it seems a little out of place to mention the statistics on the rates of all cancers in males and females between two sentences talking about breast cancer (lines 63-65 and 65-67) - unless these lines are also referring to breast cancer, in which case this needs to be written more clearly.

Reply: We have changed the order of the sentences and hopefully this is clearer now. The first sentence including males and females refers to cancer in general. 

- the statement in line 159-161 about the CONSORT diagram is a result and should be moved to the results section.

Reply: We moved this to the beginning of the results section. 

Methods

- in line 215, what you probably mean to say is that your primary measure was CBE screening uptake (perhaps more clearly, the proportion of women to took up CBE screening); the 'difference' is not an outcome but a measure of effect. Similarly, in line 219 your secondary outcomes are self-reported intention to attend CBE and mammogram screening - saying 'nature and degree...' is non-specific language in what should be unambiguous.

Reply: Thanks for helping us to clarify these points and revising the section on outcome measurement. 

- in line 220, there is need for further clarity i.e. to state explicitly which outcomes were informed by the RE-AIM framework.

Reply: Similarly, we have included now which outcomes were informed by the Re-Aim framework. 

- reading lines 232-247, it is starting to seem to me that the primary outcome of CBE uptake was a categorical (perhaps ordered categorical) variable, although I can also see how a binary outcome could have been derived out of it. And the only reason I am thinking about a binary outcome is the previous wording of how this outcome has been described and also the mention of the proportions of CBE uptake in the abstract. What I am getting at here is that the actual outcome is not clearly described anywhere. If a binary outcome was derived from the likert scale, this needs to be clearly described. If the likert scale was used as is, this also needs to be clearly described (and if this is what was done then some aspects of the results don't make sense).

Reply: The primary outcome of CBE screening uptake was a binary outcome measure (yes/no) and we have clarified this point in the methods section. A secondary outcome about reported intention to attend screening in the future was measured on a likert scale.

- if the primary outcome was binary, then although you could use chi-squared tests for a basic analysis, it would be better to use logistic regression models, if only so that you can present unadjusted and adjusted estimates of effect. In this case, chi-squared tests would be unnecessary. If the primary outcome was categorical, then this approach is fine, although you could also use multinomial regression models to obtain unadjusted and adjusted effects. If the outcome is ordered categorical, then you could use ordinal logistic regressions to obtain unadjusted and adjusted effect. Again, given the lack of clarity on exactly what the primary outcome is, I am unable to determine which of these I would recommend.

Reply: Thank you for encouraging us to think more clearly about our endpoint and the related analysis. In turn, we have consulted further with our team statistician and co-author about these important design and analysis points. The primary outcome is attendance (or not) at CBE. However, we have only self-reported data from the control group. The number/proportion of CG participants who self-reported that they had had a CBE (5/144) was very small compared to the intervention group (71/122). A logistic regression seems to be the most appropriate approach to analyse and present the data mindful of the small numbers and study limitations.

- again, for continuous outcomes, there is no need to use t-tests when you could use linear regression models to obtain unadjusted and adjusted effects. Here you don't need the change-in-screening intention score: you should simply regress the endpoint scores on treatment group (and any covariates, in the adjusted model) further adjusting for the baseline scores; this is an analysis of covariance model, which is the most statistically efficient approach for a continuous outcome with a baseline measure (and which gives identical estimates to a regression of change-from-baseline scores adjusting for baseline, given that baseline adjustment is pertinent for the improved precision of effect estimates).

Reply: We updated the analysis accordingly (Table 5). 

- the analysis exploring factors affecting screening participation is a secondary question and should not be conflated with that seeking to estimate the effect of the intervention.

Reply: We have changed the write up of the results section to focus on the effect of the intervention on the primary outcome followed by the results of the analysis of factors affecting screening participation. 

- I was not able to reproduce your sample size calculation. In my calculations, you would need about 410 participants in each arm (i.e. 820 total for two arms with 1:1 allocation) to detect a 9% absolute improvement in a binary outcome assuming a baseline level of 65%, with 80% power at the 5% level of significance for a two-tailed test. Your calculation also mentions some adjustment of recruitment to account for attrition; you would normally indicate your anticipated level of attrition as this would normally be included in the calculation

Reply: Thank you – we consulted with our team statistician and reworded the sample size calculation. 

- regarding adjustments, you would normally adjust for age (and sex if relevant, but may not be relevant here if your sample was all/mostly female) and any baseline factors relevant during randomisation.

Reply: The regression analysis adjusted for age but not sex (as all participants were women); and we assumed that randomisation would make the groups approximately comparable.

Results

- the way the results have been presented distract from the objectives of the study, in my opinion. Since the study aimed to investigate the effect of the intervention, the results should have began with a summary of the CONSORT flow diagram i.e. the numbers of participants in the treatment arms that were eligible, screened, randomised, followed-up, and for whom outcomes were available. This should have then been followed up by a table of the baseline characteristics of participants with outcomes to be included in the main analysis, split by treatment group. Crucially, this table must not include statistical tests/p-values comparing the treatment arms with respect to baseline characteristics, because in a randomised trial it is expected that the groups would be balanced and any imbalance is random and not particularly statistically interesting. Thus, the p-values in Table 3 are unnecessary, as are the separate columns for 'baseline' and 'follow-up' - this table needs to focus on the participants for whom outcomes are available and are included in the comparison of treatment groups. Tables 1 and 2 which compare characteristics of women who are included and not included are addressing a secondary question which, although interesting and even crucial in this case given the large amount of losses/attrition, does not belong to this section of the results.

Reply: Thank you - we have updated the tables and results section accordingly. 

- a results table showing the counts and proportions of participants with each binary outcome in each treatment group, along with unadjusted and adjusted effects (odds ratios) with 95% confidence intervals and p-values (and no other information, i.e. the main results) should be presented (e.g. Table 4 in https://www.sciencedirect.com/science/article/pii/S0140673618317823 ignoring the 'risk difference' column.)

Reply: We have provided this analysis. 

- (notwithstanding my comment above about the relevance of the analysis presented in Table 4) the Wald test p-values in the 5th and 7th columns for age group, ethnicity and education level should be replaced with overall (e.g. likelihood ratio) p-values, as these are the appropriate statistic to evaluate whether each variable is associated with screening attendance. The currently presented Wald test p-values only tell you which level of each variable is different from the comparison category, and they change depending on the comparison category, and don't tell you anything about the overall association between each variable and screening attendance.

Reply: We have updated this table. 

- The results in Table 5 are inappropriately presented (and this stems from the inappropriateness of the analytical approach - see my suggested analytical approach above). The appropriate way to present this would be to have the mean and standard error (not standard deviation) of each outcome measure by treatment arm, followed by the unadjusted and adjusted mean differences with 95% confidence intervals and p-values, for example Table 2 in https://www.sciencedirect.com/science/article/pii/S0140673618317823. The columns indicating the group means and SEs should indicate the numbers of individuals included in the analysis.

Reply: As above, we have updated this table.

- It is not clear which objective the analysis presented in Table 6 is addressing.

Reply: We removed this table. 

- There is a lot of attrition/losses in this trial and the number of participants eventually included in the analysis is probably smaller than what was required under the original assumptions (see my comment on the sample size calculation). A clear recognition of this should be made in the manuscript, and this is the part where some analysis of how the individuals included versus not compare (i.e. Tables 1 and 2) belong, but only at the end of the results after the analyses addressing the main objectives have been presented in a standard manner.

Reply: We changed the results section accordingly.

---

## [Decision Letter · Decision Letter 1]

18 Apr 2023

PONE-D-22-18035R1The development and evaluation of a mHealth, community education and navigation intervention to improve clinical breast examination uptake in Segamat Malaysia: a randomised control trialPLOS ONE

Dear Dr Désirée Schliemann,

Thank you for submitting your manuscript to PLOS ONE. After careful consideration, we feel that it has merit but does not fully meet PLOS ONE’s publication criteria as it currently stands. Therefore, we invite you to submit a revised version of the manuscript that addresses the points raised during the review process.

We look forward to receiving your revised manuscript.

Kind regards,

Abraham Tamirat Gizaw, PhD candidate

Academic Editor

PLOS ONE

Journal Requirements:

Additional Editor Comments:

.The qualitative part is not well  presented in the manuscript and only introduced in page 8, line 253-256. I recommend either to remove it or include method, result and discussion part of the  manuscript.

Reviewers' comments:

Reviewer's Responses to Questions

**Comments to the Author**

1. If the authors have adequately addressed your comments raised in a previous round of review and you feel that this manuscript is now acceptable for publication, you may indicate that here to bypass the “Comments to the Author” section, enter your conflict of interest statement in the “Confidential to Editor” section, and submit your "Accept" recommendation.

Reviewer #1: (No Response)

2. Is the manuscript technically sound, and do the data support the conclusions?

Reviewer #1: Partly

3. Has the statistical analysis been performed appropriately and rigorously? 

Reviewer #1: Yes

4. Have the authors made all data underlying the findings in their manuscript fully available?

Reviewer #1: No

5. Is the manuscript presented in an intelligible fashion and written in standard English?

Reviewer #1: Yes

6. Review Comments to the Author

Reviewer #1: I would like to thank the authors for the efforts in revising the manuscript. It reads much better except for a few minor areas.

1). The qualitative section has received very little attention under methods and it has a statement indicating that its results will be published separately. However, some results are provided and further discussed. Suppose a decision is made to include the qualitative results in this manuscript, then there is a need to beef up the methods section. For example, what were the interests and how was the data collected and analysed? In addition, the brief of qualitative results are about Community Health Workers (CHWs) yet the discussion under line 456 introduces readiness of nurses in the family planning clinic.

2). Line 113 is not clear and consider correcting the following typos, “used” and “lend” in lines 467 and 473 respectively.

7. PLOS authors have the option to publish the peer review history of their article (what does this mean?). If published, this will include your full peer review and any attached files.

Reviewer #1: **Yes: **BENARD OMONDI OCHIENG

---

## [Author Response · Author response to Decision Letter 1]

26 Apr 2023

Dear PLOS One Editorial Team, 

Many thanks for giving us the opportunity to revise our manuscript entitled, ‘The development and evaluation of a mHealth, community education and navigation intervention to improve clinical breast examination uptake in Segamat Malaysia: a randomised control trial’. We would like to thank the Editor and Reviewers for their comments on this manuscript. We have attempted to address all comments as outlined below and highlighted in the marked copy of the manuscript. 

If there are any further questions or suggestions for improvement, please do not hesitate to get in touch. 

Kind regards, 

Dr Desiree Schliemann

(on behalf of the study team) 

Reviewer comments:

1) The qualitative section has received very little attention under methods and it has a statement indicating that its results will be published separately. However, some results are provided and further discussed. Suppose a decision is made to include the qualitative results in this manuscript, then there is a need to beef up the methods section. For example, what were the interests and how was the data collected and analysed? In addition, the brief of qualitative results are about Community Health Workers (CHWs) yet the discussion under line 456 introduces readiness of nurses in the family planning clinic.

Response: Thank you for the feedback. We have removed the qualitative section from the manuscript as the qualitative manuscript is currently in preparation and the methods and results will be discussed there. We agree with the reviewer that it may be confusing to only include parts in this manuscript here and have therefore removed it from the methods, results and discussion. 

2) Line 113 is not clear and consider correcting the following typos, “used” and “lend” in lines 467 and 473 respectively.

Response: Thank you for highlighting this, we have corrected all three points.

---

## [Decision Letter · Decision Letter 2]

31 May 2023

PONE-D-22-18035R2The development and evaluation of a mHealth, community education and navigation intervention to improve clinical breast examination uptake in Segamat Malaysia: a randomised controlled trialPLOS ONE

Dear Dr. Schliemann,

Thank you for submitting your manuscript to PLOS ONE. After careful consideration, we feel that it has merit but does not fully meet PLOS ONE’s publication criteria as it currently stands. Therefore, we invite you to submit a revised version of the manuscript that addresses the points raised during the review process.

We look forward to receiving your revised manuscript.

Kind regards,

Abraham Tamirat Gizaw, PhD candidate

Academic Editor

PLOS ONE

Journal Requirements:

Additional Editor Comments:

Pleased clarify the sample size calculation in details.

Reviewers' comments:

Reviewer's Responses to Questions

**Comments to the Author**

1. If the authors have adequately addressed your comments raised in a previous round of review and you feel that this manuscript is now acceptable for publication, you may indicate that here to bypass the “Comments to the Author” section, enter your conflict of interest statement in the “Confidential to Editor” section, and submit your "Accept" recommendation.

Reviewer #2: (No Response)

2. Is the manuscript technically sound, and do the data support the conclusions?

Reviewer #2: Yes

3. Has the statistical analysis been performed appropriately and rigorously? 

Reviewer #2: Yes

4. Have the authors made all data underlying the findings in their manuscript fully available?

Reviewer #2: No

5. Is the manuscript presented in an intelligible fashion and written in standard English?

Reviewer #2: Yes

6. Review Comments to the Author

Reviewer #2: The authors have responded appropriately to the previous round of comments and the manuscript has clearly been improved.

A few lingering issues:

- the sample size calculation is still not clearly reported; it seems to suggest a total sample size of 932 for both 80% power and 90% power, which isn't right. Also, the authors indicate that the sample size was adjusted for losses (attrition, etc). You need to specify what adjustment was made for losses, e.g. 10%? The statement also needs to be clear about whether this calculation refers to the total number of individuals required, or the number of individuals required per group.

- still on sample size, the statement reporting the number of women randomised (line 261: "IG (n=888 or CG (n=881)" is a result and belongs to the results section, if not already reported there. IT should be removed from here.

- the results now seem well-reported; to pay attention to bits such as in line 311 and 315 where a character (I believe the 'chi' character) is missing.

7. PLOS authors have the option to publish the peer review history of their article (what does this mean?). If published, this will include your full peer review and any attached files.

Reviewer #2: No

---

## [Author Response · Author response to Decision Letter 2]

6 Jun 2023

Dear PLOS One Editorial Team, 

Many thanks for giving us the opportunity to revise our manuscript entitled, ‘The development and evaluation of a mHealth, community education and navigation intervention to improve clinical breast examination uptake in Segamat Malaysia: a randomised control trial’. We would like to thank the Editor and Reviewers for their comments on this manuscript. We have attempted to address all comments as outlined below and highlighted in the marked copy of the manuscript. 

If there are any further questions or suggestions for improvement, please do not hesitate to get in touch. 

Kind regards, 

Dr Desiree Schliemann

(on behalf of the study team)

Journal Requirements:

Response: We have revised the reference list and removed any duplicate references. We did not include any retracted articles. 

Additional Editor Comments:

Pleased clarify the sample size calculation in details.

A few lingering issues:

- the sample size calculation is still not clearly reported; it seems to suggest a total sample size of 932 for both 80% power and 90% power, which isn't right. Also, the authors indicate that the sample size was adjusted for losses (attrition, etc). You need to specify what adjustment was made for losses, e.g. 10%? The statement also needs to be clear about whether this calculation refers to the total number of individuals required, or the number of individuals required per group.

Response: We have updated the sample size calculation. 

- still on sample size, the statement reporting the number of women randomised (line 261: "IG (n=888 or CG (n=881)" is a result and belongs to the results section, if not already reported there. IT should be removed from here.

Response: We moved this from the methods into the results section.

- the results now seem well-reported; to pay attention to bits such as in line 311 and 315 where a character (I believe the 'chi' character) is missing.

Response: We have included the ꭓ (Chi-squared) symbol in the Word document but it may have gotten lost when the document was converted into a PDF document. Hopefully the editorial team is able to include the chi-squared symbol from the Word documents in the final publication.

---

## [Editor Report · Decision Letter 3]

28 Jun 2023

The development and evaluation of a mHealth, community education and navigation intervention to improve clinical breast examination uptake in Segamat Malaysia: a randomised controlled trial

PONE-D-22-18035R3

Dear Dr.Desiree Schliemann

,We’re pleased to inform you that your manuscript has been judged scientifically suitable for publication and will be formally accepted for publication once it meets all outstanding technical requirements.

Kind regards,

Abraham Tamirat Gizaw, PhD candidate

Academic Editor

PLOS ONE

Additional Editor Comments (optional):

Reviewers' comments:

<quillbot-extension-portal></quillbot-extension-portal>

---

## [Editor Report · Acceptance letter]

26 Sep 2023

PONE-D-22-18035R3 

The development and evaluation of a mHealth, community education and navigation intervention to improve clinical breast examination uptake in Segamat Malaysia: a randomised controlled trial 

Dear Dr. Donnelly:

I'm pleased to inform you that your manuscript has been deemed suitable for publication in PLOS ONE. Congratulations! Your manuscript is now with our production department. 

Kind regards, 

on behalf of

Asst.Professor Abraham Tamirat Gizaw 

Academic Editor

PLOS ONE